# Real-Life Experience on Dolutegravir and Lamivudine as Initial or Switch Therapy in a Silver Population Living with HIV

**DOI:** 10.3390/v15081740

**Published:** 2023-08-15

**Authors:** Maria Mazzitelli, Lolita Sasset, Samuele Gardin, Davide Leoni, Mattia Trunfio, Vincenzo Scaglione, Daniele Mengato, Elena Agostini, Eleonora Vania, Cristina Putaggio, Annamaria Cattelan

**Affiliations:** 1Infectious and Tropical Diseases Unit, Padua University Hospital, 35128 Padua, Italy; 2Infectious Diseases Unit, Department of Medical Sciences, University of Turin, Amedeo di Savoia Hospital, 10149 Turin, Italy; mattia.trunfio@edu.unito.it; 3HIV Neurobehavioral Research Program, Departments of Neurosciences and Psychiatry, School of Medicine, University of California, San Diego, CA 92093, USA; 4Hospital Pharmacy Unit, Padua University Hospital, 35128 Padua, Italy; 5Infectious Disease Unit, Department of Medicine, University of Udine, 33100 Udine, Italy; 6Department of Molecular Medicine, University of Padua, 35131 Padua, Italy

**Keywords:** HIV, PLWH, dual therapy, elderly, dolutegravir/lamivudine, durability

## Abstract

Background: Clinical trials and real-life studies have granted the efficacy and safety of dolutegravir and lamivudine (DTG/3TC) in naïve and experienced people living with HIV (PLWH), but there are no long-term data in elderly people. We herein describe our real-life cohort of PLWH who were ≥65 years of age (PLWH ≥ 65) who started or were switched to DTG/3TC, single-tablet regimen, or DTG plus 3TC. Methods: We considered laboratory/clinical parameter changes from the baseline to the last follow-up time point available for each person by the paired Wilcoxon test and analyzed factors associated with virological failure (VF) and discontinuation. Results: We included 112 PLWH with a median age of 66 (IQR: 65–70) years, 77.6% males; 84.8% of people had multimorbidity, 34.8% were on polypharmacy, and only 5.4% were naïve to treatment. Reasons to be switched to DTG/3TC were: abacavir removal (38.7%), treatment simplification (33.1%), and PI discontinuation (28.2%). The median treatment durability was 6 (IQR: 5.4–7) years. No significant changes were detected in metabolic, renal, immunological, or cardiovascular biomarkers during follow-up. HIV RNA undetectability was maintained in 104 (92.8%) individuals for whom follow-up evaluation was available. We observed eight discontinuations (two deaths, two VFs, two early intolerances, one significant weight gain, and one switch to long-acting therapy). No factors were significantly associated with VF or discontinuation. Conclusions: This is the first study on DTG/3TC in PLWH ≥ 65 with a follow-up longer than 5 years. DTG/3TC was found to be safe and effective, neutral on metabolic parameters, and with a low discontinuation rate for toxicity or VF.

## 1. Introduction

The pivotal moment in the evolution of antiretroviral therapy came in the mid-1990s with the advent of highly active antiretroviral therapy (HAART), a combination of multiple antiretroviral drugs from different classes to target various stages of the HIV lifecycle simultaneously [1,2]. HAART marked a turning point in the management of HIV/AIDS. It led to a dramatic reduction in mortality rates and opportunistic infections, effectively transforming HIV from a fatal illness into a manageable chronic condition for many individuals [1,2,3]. However, HAART was not without challenges. The complex dosing schedules, potential for drug interactions, and adverse side effects posed obstacles to optimal adherence. As our understanding of HIV and antiretroviral drugs deepened, researchers focused on developing more convenient and tolerable treatment regimens. This led to the development of fixed-dose combination pills, containing multiple drugs in a single tablet. These advancements not only improved adherence but also reduced the pill burden and simplified treatment regimens.

Consequently, in recent decades we have been witnessing the progressive aging of people living with HIV (PLWH) [3,4]. Furthermore, a significant proportion of new diagnoses occur in patients who are already advanced in age or over 50 years of age [5,6]. Clinical challenges in this population, however, remain, such as managing the increasing burden of comorbidities, how to manage polypharmacy and drug interactions, and how to prevent toxicity, including that caused by antiretroviral therapy. This complexity has led clinicians to implement dedicated services for the elderly aging with HIV to guarantee a holistic approach and optimal care delivery [5,7,8]. The main comorbidities of the elderly are cardiovascular disease, the progressive decline of renal function, and osteoporosis; therefore, it is increasingly necessary to identify antiretroviral regimens that ideally do not modify or aggravate these burdens or even allow them to be lightened [9,10,11]. One of the latest developments in HIV treatment is the emergence of dual-therapy regimens, a strategic approach aimed at maintaining viral suppression while potentially reducing the pill burden and minimizing long-term side effects associated with traditional triple-drug therapies. In this context, the combination of dolutegravir and lamivudine has garnered significant attention. Dolutegravir, an integrase strand transfer inhibitor (INSTI), and lamivudine, a nucleoside reverse transcriptase inhibitor (NRTI), have each demonstrated potent antiretroviral activity and a favorable safety profile. Research studies, such as the landmark GEMINI trials (GEMINI-1 and GEMINI-2), have explored the efficacy and safety of dolutegravir–lamivudine dual therapy compared to standard triple-drug regimens [12,13]. These trials found that the dual-therapy approach was noninferior to traditional triple therapy in achieving and maintaining viral suppression among treatment-naïve individuals with HIV-1 infection [12,13]. Furthermore, in a large randomized clinical trial, the switch to DTG plus 3TC or FTC was noninferior to continuing three-drug therapy in individuals with virologic suppression and no history of prior virologic failure or resistance to these agents [14]. Therefore, a regimen of dolutegravir and lamivudine received approval for HIV-1 treatment in Europe in 2019, and it is currently recommended by international guidelines [15,16]. Furthermore, the simplified dual-therapy regimen offers the potential for reduced toxicity and metabolic complications, which are often associated with long-term use of certain antiretroviral agents.

Randomized and controlled clinical trials, as mentioned, but also real-life data have in recent years confirmed and sealed the efficacy of the dolutegravir (DTG) and lamivudine (3TC) regimen both in therapy switching, but also in naïve PLWH [17,18]. As the landscape of HIV treatment continues to evolve, it is essential to critically examine the clinical implications and long-term outcomes of novel therapeutic approaches, especially in special patient populations. We therefore performed a single-center retrospective study to assess the long-term viro-immunological and clinical outcomes of elderly HIV-positive people receiving a dual-antiretroviral regimen with dolutegravir plus lamivudine.

## 2. Materials and Methods

This is a single-center, retrospective, observational study conducted at the Infectious and Tropical Diseases Unit of Padua University Hospital (Northern Italy). It complies with principles of good clinical practice and was conducted in accordance with Declaration of Helsinki. The study protocol was provided to the Ethical Committee (study protocol n. 5285/AO). Participants’ consent was waived as per Italian law [19]. We included all PLWH who were ≥65 years of age and had ever started a cART containing DTG and 3TC, either in a single- or in a dual-tablet combination, in our center (the date in which the first person with HIV started this regimen in our center was 8 April 2015). Follow-up was closed on 31 May 2023. Data were retrieved from medical health records and collected in an anonymized Excel^®^ dataset. We included demographics (age, sex, and ethnicity), HIV-related characteristics (risk factors for HIV acquisition, years living with HIV, CD4+ T cell count at nadir, type of cART before switch in experienced PLWH, and previous AIDS episodes in medical history), laboratory findings (HIV RNA, CD4+ T cell count, CD4/CD8 ratio, total cholesterol, HDL, LDL, triglycerides, serum creatinine, and glucose), and anthropometric parameters (weight, height, and waist circumference). We categorized antiretroviral regimens at the baseline as: naïve (no antiretroviral treatment); mono/dual regimen (including PLWH who were on protease inhibitor monotherapy or on dual regimens based on protease inhibitor combinations); 2 nucleos(t)ide reverse transcriptase inhibitors plus a protease inhibitor; 2 nucleos(t)ide reverse transcriptase inhibitors plus a non-nucleos(t)ide reverse transcriptase inhibitor; and 2 nucleos(t)ide reverse transcriptase inhibitors plus an integrase inhibitor. Reasons for switching to DTG/3TC were classified as: abacavir discontinuation (to reduce cardiovascular risk), protease inhibitor discontinuation (occurred due to dyslipidemia or drug–drug interactions), and simplification (to reduce pill burden or number of antiretrovirals).

We also included a list of the following comorbidities: malignancy, chronic renal disease (defined as an estimated glomerular filtration rate of <60 mL/min) [20], dyslipidemia, ischemic heart disease, hypertension, obesity (defined as a body mass index of >30 kg/m^2^) [21], cirrhosis, diabetes, osteoporosis, chronic obstructive pulmonary diseases, mood disorders, and neurological disorders. For each participant, we considered the median number of comorbidities and recorded the prevalence of multimorbidity (defined as the presence of 2 or more noninfectious comorbidities in the same person) and polypharmacy (defined as the intake of 5 or more non-antiretroviral medications in the same person) [7,22,23,24,25].

We recorded changes of any parameters from the baseline to the last follow-up time point available for each person and analyzed factors associated with virological failure (VF) and discontinuation. Continuous and categorical variables were reported as median values (interquartile range) and absolute numbers (proportion), respectively. The intra-subject median changes from baseline to follow-up were analyzed by the paired Wilcoxon test after excluding subjects classified as discontinuations. Binary and multinomial logistic regression was considered for univariate *p*-values < 0.1, the value set-up for statistical significance. The analyses were run through Stata SE 16 (Stata Corp LLC, College Station, TX, USA). Data are expressed as medians (interquartile range) or percentages, as appropriate. 

## 3. Results

From 8 April 2015 to 31 May 2023 (date of the last follow-up time point), 112 PLWH started DTG and 3TC in our center. In Table 1 we summarized all the cohort characteristics. The median age was 66 years (IQR: 65–70), and 87/112 (77.6%) were male at birth; 97.3% PLWH were of Caucasian ethnicity. The median time living with HIV was 25 (20–29) years. The median CD4+ T cell count at nadir was 233 (122–345) cells/mm^3^, while 24.1% of PLWH had experienced at least one AIDS-defining condition according to their past medical history. Six PLWH (5.4%) started this combination as a first-line regimen, while the remaining ones were switched off from other antiretrovirals. For these people, the zenith of HIV RNA was 74.100 (IQR: 63.000–86.700) copies/mL, and the median CD4+ T cell count at HIV diagnosis time was 451 (IQR: 240–748) cell/mm^3^. As for people who switched, HIV was very well controlled; 100% of such persons had HIV RNA levels of <50 copies/mL, and the median CD4+ T cell count was 645 (466–872) cell/mm^3^. The median duration of HIV infection was 25 (20–29) years, and the median CD4+ T cell count at nadir was 233 cell/mm^3^. Almost one out of four (24.1%) individuals had experienced an AIDS event in their past medical history. The vast majority of PLWH in our cohort presented multimorbidity (84.8%), while polypharmacy was present in 34.8% of cases. Each participant experienced a median number of four comorbidities (IQR: 2–6). The most common comorbidities were hypertension (50.9%), dyslipidemia (48.2%), osteoporosis (26.8%), cancer (24.1%), and chronic kidney disease (24.2%). Almost one of four (23.2%) and one of five (21.4%) PLWH in our cohort were affected by mood and neurological disorders, respectively. The most common reason to be switched to DTG+3TC was the removal of abacavir from the previous regimen (38.7%) due to high cardiovascular risk, followed by simplification (33.1%) and PI discontinuation (28.2%) due to high cardiovascular risk and metabolic toxicities. In addition, 47% of PLWH started this combination as a two-pill regimen, and all but six were switched to a single-tablet regimen as soon as it was available.

The median durability of follow-up of the DTG+3TC regimen was 6.0 (5.4–7.0) years. Along this period, we observed eight discontinuations: two participants who were deceased at weeks 70 and 95, respectively, for reasons not related to the antiretroviral regimen; two viral failures which occurred in two experienced people with no resistance selection at weeks 74 and 77; two interruptions which occurred due to neuropsychiatric intolerances at weeks 1 and 2; one due to significant weight gain (12 kg) at week 66; and one due to a person choosing to be switched to long-acting treatment at week 90. Seven discontinuations occurred in experienced people, while one occurred in a naïve person. One of the two cases of neuropsychiatric toxicity occurred in a naïve person. No clinical or laboratory parameters were associated with discontinuation. 

Table 2 shows the evolution of the study parameters from baseline to the last follow-up time point available for each PLWH. We did not observe any significant changes in total cholesterol, HDL, LDL, triglycerides, or glucose levels. Kidney function, estimated by eGFR, did not show any significant decline. Liver function tests (AST and ALT) remained within the normal range and did not have any significant increase. BMI remained stable over the follow-up. As for the different scores, the Framingham, NAFLD, and APRI scores remained substantially stable, with no significant changes, while we observed a statistically significant reduction in the FIB-4 score, which, however, was not clinically relevant. While as expected, we observed a significant change both in viral loads (all people reached HIV RNA undetectability) and in immunological parameters (CD4+ T cell count and CD4/CD8 ratio) in PLWH who were naïve when DTG+3TC was initiated, no relevant changes were observed in people who had been switched to this new regimen. 

## 4. Discussion

The use of cART in geriatric PLWH, defined as people aged 65 years and older, presents unique challenges and considerations due to the intersection of HIV infection and the complexities of aging. As the aging population living with HIV grows, optimizing treatment approaches for this specific population becomes increasingly important. Several studies have shed light on the benefits, challenges, and evolving strategies related to antiretroviral use in geriatric patients [7,23,24,26]. The most recognized benefits are the improved survival and quality of life, with mitigation of disease progression and related complications, better maintenance of immunological function, and reduced risk of opportunistic infections [1,2,3,4]. By contrast, some challenges remain. Geriatric PLWH often have multiple comorbidities and may be taking multiple medications [7,24,26,27,28,29]. Antiretroviral selection must consider potential drug–drug interactions and the impact on pre-existing health conditions [26,27]. In addition, older individuals may be more susceptible to drug-related toxicities and adverse effects; hence, careful selection of ART regimens that balance efficacy with tolerability is crucial. Indeed, physiological changes associated with aging can affect drug absorption, distribution, metabolism, and excretion. Antiretroviral dosing and pharmacokinetics may need adjustment, especially with physiological or pathological decline of kidney function, bone mineral density reduction, and increased cardiovascular risk [7,24,26,27,28,29]. Finally, geriatric PLWH may face social isolation and mental health disturbances, impacting importantly quality of life and treatment adherence. Holistic care that addresses all these issues contributes to the global well-being of aging HIV-positive people [7,26,27,28,29].

Our real-life observational study showed that the combination of DTG and 3TC was effective and safe in elderly PLWH in the long-term follow-up. To our knowledge, it is the first report that documented such results in PLWH who are ≥65 years of age during a median follow-up time of 6 years.

With advances in antiretroviral therapy and more potent, convenient, and tolerable regimens, virological suppression is easily achieved in most PLWH, and dual therapies have indeed been a major evolution over the triple-antiretroviral ones. 

Data are emerging on the safety and efficacy of other modern dual therapies based on INIs, such as dolutegravir and rilpivirine (RPV) or dolutegravir and doravirine (DOR) as switch strategies in people who have already gained virological suppression [30,31,32,33].

The SWORD-1 and SWORD-2 studies granted the efficacy, tolerability and safety, and noninferiority of DTG/RPV compared to the standard triple therapy in maintaining virological suppression in PLWH who switch their regimen [30]. Regarding DTG/DOR, we do not at this time have randomized clinical trials, but evidence has come mostly from observational and real-life cohorts [31,32,33]. In comparison with DTG/RPV and DTG/3TC, and despite this regimen not being mentioned in the guidelines, DTG/DOR is used mostly in highly treated PLWH to build a maintenance therapy with no drug–drug interactions and with a high genetic barrier [34]. However, DTG/3TC remains to date the only dual treatment approved by guidelines both in naïve and experienced PLWH, and with the longest follow-up and the most robust data [12,13,14,15,16,17,18]. Even though first-line recommended regimens consist of two nucleos(t)ide reverse transcriptase inhibitors plus an integrase inhibitor, dual therapy including DTG and 3TC is nowadays also considered as a recommended initial regimen in naïve PLWH by all international guidelines [15,16]. In addition, DTG + 3TC represents a well-studied switching strategy in virologically suppressed people. Our real-life data in a geriatric population confirmed that both among people who switched and among naïve PLWH and for whom a follow-up assessment was available, more than 92% of people maintained virological suppression (HIV RNA < 20 copies/mL). Both the CD4+ T cell count and the CD4/CD8 ratio were maintained during the whole study period, with a slight significant increase in the CD4/CD8 ratio in experienced patients. These results confirmed the high rate of viral suppression documented in the 144-week secondary analysis [35] of the GEMINI-1 and GEMINI-2 studies [12,13] and in a recent real-world study of switching to DTG/3TC during a three-year follow-up [36]. However, these studies were conducted in younger adult subjects, also suggesting no difference in virologic response between older and younger patients. 

In our cohort, two persons experienced virological failure, but no one developed treatment-emergent resistance mutations at the time of virological failure, as already reported by a recent metanalysis [37]. This allowed us the choice of simple further antiretroviral regimens without using boosted protease inhibitors. In these two persons, virological failure occurred due to lack of adherence to antiretroviral treatment. Indeed, they disclosed during the visits detecting the virological failures a very suboptimal treatment intake (i.e., less than 60% and 50% doses in each one of them, respectively).

Beyond maintaining virological control, the main focus today in PLWH is to achieve and preserve well-being and good quality of life. This is particularly crucial in the aging population, whose prevalence is constantly increasing and will be higher than 70% in 2030 [38]. Moreover, even though most clinical trials on antiretrovirals have tried to include aging populations, the proportion of elderly enrolled has been very limited, especially in studies considering the naïve population. In combined data from six trials in treatment-naïve and treatment-experienced participants receiving DTG- or non-DTG-based regimens, individuals who were ≥65 years old were numerically very few, numbering 23 and 14, respectively [39]. In the TANGO trial, only 21% to 25% of PLWH enrolled in each arm were at least 50 years of age, showing that older patients are underrepresented in clinical trials. A recent study reported that switching to bictegravir/tenofovir alafenamide/emtricitabine was effective and well-tolerated in 86 virologically suppressed adults aged >65 years; however, the length of follow-up was limited to 96 weeks [40].

By aging, PLWH experience a longer period of life characterized by multimorbidity, with complex drug regimens that will continue to be a challenge for clinical management, due to possible side effects, toxicity, and drug interactions [8,11,24]. In our study, almost 85% of persons presented multimorbidity, and polypharmacy was present in 35% of cases. The proportion of people experiencing multimorbidity and the distributions of the different comorbidities in our population are substantially similar to those reported in other geriatric cohorts, where dyslipidemia and hypertension are the most commonly reported comorbidities [23,41,42,43]. The slight differences in terms of numbers in the different cohorts may be explained by differences in lifestyle, socioeconomic status, and environmental factors.

Multimorbidity and polypharmacy are of crucial concern for aging PLWH, whether they are antiretroviral-naïve or antiretroviral-experienced. Multimorbidity may drive and influence the choice of antiretroviral regimen and lead clinicians to select regimens that are both free from cardiovascular/kidney toxicity and metabolic/bone-friendly. In our study, the most important reason for switching to DTG+3TC was the removal of abacavir from the previous regimen due to high cardiovascular risk; the removal of protease inhibitors was also present in almost 30% of cases. Furthermore, multiple comorbidities lead to polypharmacy and render HIV-positive elderly people vulnerable to medication-related problems, including adverse drug reactions and potential drug–drug interactions [24]. During the long-term follow-up of our study, no serious adverse events or significant changes in metabolic parameters were recorded, showing the safety profile of the DTG/3TC regimen, which is consistent with previously published data [17,18,44,45]. Of note, only one patient experienced significant weight gain, which was the reason for their discontinuation of the regimen at week 66. These data are reassuring, considering that among INIs, DTG is the most involved in weight gain [46]. However, we would underline that our cohort of patients involved predominantly white males, whereas integrase inhibitor-related weight gain was more frequently observed in females and those from a black African background among PLWH [47,48]. 

Two persons in our study discontinued the DTG/3TC regimen early because of neuropsychiatric toxicities, which began soon after the regimen initiation. 

This area is of growing concern, as it is nowadays known that the emergence of neuropsychiatric toxicities is associated with the use of INIs, raising important questions about their safety and clinical implications over time [49,50,51]. Neuropsychiatric toxicities encompass a range of neurological and psychiatric symptoms that can affect cognition, mood, behavior, and overall mental well-being. While the mechanisms underlying these toxicities are still being elucidated, it is increasingly recognized that certain INIs, primarily dolutegravir, have been linked to a higher incidence of neuropsychiatric adverse events, such as sleep disturbances, vivid dreams, depression, and anxiety [49,50,51].

On this topic, it has been observed that in geriatric PLWH, dolutegravir maximal concentrations were increased by 25%, even if this increase was not correlated with modification of sleep function [52]. However, some authors feel that there is a need for more pharmacokinetic data in elderly PLWH, especially those with comorbidities or frailty [52].

This study is somewhat limited by the low number of participants and by its retrospective design. With a small sample size, it becomes challenging to generalize the findings to a larger population. The characteristics of the patients (mostly males and MSM) in the study might not be representative of the broader patient population, making it difficult to apply the results to other settings or patient groups (females, people from ethnicities other than Caucasian). It is crucial to include in such kinds of studies a larger proportion of post-menopausal women. Lastly, we did not include a control group following other antiretroviral regimens, whether dual or triple ones. 

## 5. Conclusions

Finally, even if longitudinal studies including a larger number of PLWH are needed to confirm our results, in this retrospective observational study the dual regimen of DTG/3TC has demonstrated its effectiveness in a geriatric cohort of PLWH in a real-world setting. The high rate of virological response, the good safety profile, and the low rate of drug–drug interactions together with the reduction in the daily number of antiretroviral drugs taken by PLWH makes this regimen particularly favorable for individuals with comorbidities and polypharmacy. It is worth noting how the use of antiretroviral therapy in geriatric patients requires a nuanced approach that considers both HIV-related factors and age-related considerations. Therefore, tailoring treatment regimens to the specific needs of geriatric patients is crucial for promoting healthy aging and improving their overall quality of life.

## Figures and Tables

**Table 1 viruses-15-01740-t001:** Cohort description.

Characteristics	Study Population (*n* = 112)
Age, years, median (IQR)	66 (65–70)
Male gender, *n* (%)	87 (77.6)
White race, *n* (%)	109 (97.3)
Length of HIV infection, years, median (IQR)	25 (20–29)
CD4+ T cell count nadir, cells/mm^3^	233 (122–345)
Past AIDS episodes, *n* (%)	27 (24.1)
HCV coinfection, *n* (%)	21 (18.7)
HIV acquisition route, *n* (%)	
MSM	61 (54.5)
Heterosexual	34 (30.3)
IVDU	15 (13.4)
Blood products	2 (1.8)
Previous antiretroviral, *n* (%)	
Naïve	6 (5.4)
Mono/dual regimen	25 (22.3)
2NRTI+PI	4 (3.6)
2NRTI+NNRTI	11 (9.8)
2NRTI+INSTI	58 (51.8)
Reason to switch, *n* (%)	
ABC discontinuation	41 (38.7)
PI discontinuation	22 (28.2)
Simplification	35 (33.1)
Comorbidities, *n* (%)	
Malignancy	27 (24.1)
Chronic renal disease	27 (24.1)
Dyslipidemia	54 (48.2)
Ischemic heart disease	19 (17.0)
Hypertension	57 (50.9)
Obesity	25 (22.3)
Cirrhosis	3 (2.7)
Diabetes	21 (18.8)
Osteoporosis	30 (26.8)
COPD	12 (10.7)
Mood disorders	26 (23.2)
Neurological disorders	24 (21.4)
N of comorbidities per participant, median (IQR)	4 (2–6)
Multimorbidity, *n* (%)	95 (84.8)
N of comedications per participant, median (IQR)	4 (2–5)
Polypharmacy, *n* (%)	39 (34.8%)

Legend of Table 1: IQR = interquartile range, MSM = men who have sex with men, IVDU = intravenous drug use, NRTI = nucleos(t)ide reverse transcriptase inhibitors, PI = protease inhibitors, NNRTI = non nucleos(t)ide reverse transcriptase inhibitors, INSTI = integrase inhibitors, ABC = abacavir, COPD = chronic obstructive pulmonary disease, N = number.

**Table 2 viruses-15-01740-t002:** Evolution of the study parameters.

Parameter	Baseline (*n* = 104)	End of Follow-Up (*n* = 104)	*p*
HIV-RNA, cp/mL, median (IQR)	0 (0–0)	0 (0–0)	-
HIV-RNA *, cp/mL, median (IQR)	74.100 (63.000–86.700)	0 (0–0)	-
CD4+ T cell count, cells/mmc, median (IQR)	663 (419–775)	614 (490–799)	0.537
CD4+ T cell count, cells/mmc *, median (IQR)	451 (240–748)	780 (625–836)	0.042
CD4/CD8 ratio, median (IQR)	0.91 (0.60–1.31)	0.92 (0.52–1.40)	0.632
CD4/CD8 ratio *, median (IQR)	0.5 (0.47–0.48)	1.1 (0.91–1.5)	0.003
Total cholesterol, mmol/dL, median (IQR)	4.8 (4.0–5.4)	4.7 (3.7–5.3)	0.091
HDL, mmol/dL, median (IQR)	1.3 (1.1–1.6)	1.2 (1.1–1.6)	0.957
LDL, mmol/dL, median (IQR)	2.9 (2.3–3.7)	2.9 (2.3–3.6)	0.412
Triglycerides, mmol/dL, median (IQR)	1.2 (0.91–1.8)	1.2 (0.82–1.6)	0.069
Glucose, mmol/dL, median (IQR)	5.4 (4.9–6.1)	5.3 (4.7–6.2)	0.239
eGFR, mL/min, median (IQR)	74 (62–89)	73 (60–88)	0.105
ALT, IU/mL, median (IQR)	23 (17–29)	22 (17–31)	0.700
AST, IU/mL, median (IQR)	24 (17–30)	25 (20–31)	0.609
Albumin, mmol/dL, median (IQR)	41 (39–44)	40 (39–43)	0.638
NAFLD, median (IQR)	−0.84 (−1.32; −0.075)	−0.73 (−1.41; −0.0097)	0.110
Framingham, median (IQR)	30.0 (18.4–30.0)	30.0 (18.4–30.0)	0.673
FIB-4, median (IQR)	0.036 (0.017–0.56)	0 (0–0.0086)	<0.001
APRI score, median (IQR)	0.31 (0.24–0.44)	0.31 (0.21–0.44)	0.877
BMI, median (IQR)	25.9 (24.0–29.6)	25.9 (23.9–29.4)	0.360

* ART-naïve participants. Legend to Table 2: IQR = interquartile range, FIB-4 = Fibrosis-4 Index, APRI = AST-to-platelet ratio index, NAFLD = Non-alcoholic fatty liver disease, BMI = body mass index.

## Data Availability

The data that support the findings of this study are available on request from the corresponding author. The data are not publicly available due to privacy or ethical restrictions.

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
