# Peer review of "Real-Life Experience on Dolutegravir and Lamivudine as Initial or Switch Therapy in a Silver Population Living with HIV"

_viruses, 2023, doi:10.3390/v15081740_

Round 1

Reviewer 1 Report

This study proves to be valuable as it specifically targets a real-life cohort of individuals aged ≥65 living with HIV (PLWH≥65) who initiated or transitioned to DTG/3TC. The authors tracked the evolution of study parameters from baseline to the last available follow-up for each PLWH and demonstrated that DTG/3TC treatment is both safe and effective. Notably, the medication showed a neutral impact on metabolic parameters and exhibited a low discontinuation rate due to toxicity or virologic failure (VF).

Below are my comments:

1.     In the manuscript, the author claimed that "we can see a significant change in viral load and immunological parameters in PLWH who were naïve when DTG+3TC was initiated." However, I'm unable to ascertain how this conclusion was reached based on table 2. Could the authors specify the exact parameters they are referring to and provide the corresponding p-values for these changes?

2.     I noticed that the baseline for the first two parameters is reported as zero. Could you please clarify the rationale behind this?

3.     In the discussion, the author mentioned that "no PLWH developed treatment emergent resistance mutations at the time of virological failure, allowing the choice of simple further antiretroviral regimens." If the virological failure was not caused by resistance mutations, could you please clarify what factors were responsible for it?

To enhance the quality of their manuscript, the authors need to consider refining their use of English. 

Author Response

Reviewer #1

This study proves to be valuable as it specifically targets a real-life cohort of individuals aged ≥65 living with HIV (PLWH≥65) who initiated or transitioned to DTG/3TC. The authors tracked the evolution of study parameters from baseline to the last available follow-up for each PLWH and demonstrated that DTG/3TC treatment is both safe and effective. Notably, the medication showed a neutral impact on metabolic parameters and exhibited a low discontinuation rate due to toxicity or virologic failure (VF).

Dear reviewer, thank for your time and your comments, we made all the changes and amendments you suggested.

Below are my comments:

  1. In the manuscript, the author claimed that "we can see a significant change in viral load and immunological parameters in PLWH who were naïve when DTG+3TC was initiated." However, I'm unable to ascertain how this conclusion was reached based on table 2. Could the authors specify the exact parameters they are referring to and provide the corresponding p-values for these changes?
  2. I noticed that the baseline for the first two parameters is reported as zero. Could you please clarify the rationale behind this?

Thanks for your comments. Table 2 was amended, by adding both rows with median values for naïve and experienced PLWH. We feel that now it is clearer. In the first version there were some mistakes.

  1. In the discussion, the author mentioned that "no PLWH developed treatment emergent resistance mutations at the time of virological failure, allowing the choice of simple further antiretroviral regimens." If the virological failure was not caused by resistance mutations, could you please clarify what factors were responsible for it?

Thanks for raising this point. The reason of virological failure were low level od adherence and suboptimal antiretroviral intake (please see lines 250-256).

  1. To enhance the quality of their manuscript, the authors need to consider refining their use of English.

English has been reviewed and many typos amended.

Reviewer 2 Report

This retrospective study on long-term follow-up of DTG/3TC combination in older PLWH is of interest to HIV/AIDS clinicians.  It provides reassuring information, it is well written, all sections are clear and concise.  Except for the retrospective nature, I see no methodological weaknesses.

The English language is fine, but there are a few very minor corrections to be made:

-          P. 2: …sealed the efficacy of the dolutegravir (DTG) and lamivudine (3TC) regimen both in therapy switching, but also in naïve [12, 13] : PLWH to be added

-          P. 2: As for people who switch, HIV was very well controlled: SWITCHED

-          P. 5: It is particularly crucial in aging population, whose prevalence it constantly increasing and it is estimated to overcome the 70% in 2030 [17].  Prevalence  IS constantly increasing and WILL BE MORE  THAN 70 % in 2030

Author Response

Reviewer #2

This retrospective study on long-term follow-up of DTG/3TC combination in older PLWH is of interest to HIV/AIDS clinicians.  It provides reassuring information, it is well written, all sections are clear and concise.  Except for the retrospective nature, I see no methodological weaknesses. 

 Dear reviewer, thank for your time and your comments, we made all the changes and amendments you suggested.

Comments on the Quality of English Language

The English language is fine, but there are a few very minor corrections to be made:

-          P. 2: …sealed the efficacy of the dolutegravir (DTG) and lamivudine (3TC) regimen both in therapy switching, but also in naïve [12, 13] : PLWH to be added

-          P. 2: As for people who switch, HIV was very well controlled: SWITCHED

-          P. 5: It is particularly crucial in aging population, whose prevalence it constantly increasing and it is estimated to overcome the 70% in 2030 [17].  Prevalence  IS constantly increasing and WILL BE MORE  THAN 70 % in 2030

All these points and other typos have been amended.